# Automatically Detecting Incoherent Written Math Answers of Fourth-Graders

Felipe Urrutia [†] , Roberto Araya *,[†]

Centro de Investigación Avanzada en Educación, Instituto de Educación, Universidad de Chile, Santiago 8320000, Chile; furrutia@dim.uchile.cl
* Correspondence: roberto.araya@ciae.uchile.cl
† These authors contributed equally to this work.

**Abstract:** Arguing and communicating are basic skills in the mathematics curriculum. Making arguments in written form facilitates rigorous reasoning. It allows peers to review arguments, and to receive feedback about them. Even though it requires additional cognitive effort in the calculation process, it enhances long-term retention and facilitates deeper understanding. However, developing these competencies in elementary school classrooms is a great challenge. It requires at least two conditions: all students write and all receive immediate feedback. One solution is to use online platforms. However, this is very demanding for the teacher. The teacher must review 30 answers in real time. To facilitate the revision, it is necessary to automatize the detection of incoherent responses. Thus, the teacher can immediately seek to correct them. In this work, we analyzed 14,457 responses to open-ended questions written by 974 fourth graders on the ConectaIdeas online platform. A total of 13% of the answers were incoherent. Using natural language processing and machine learning algorithms, we built an automatic classifier. Then, we tested the classifier on an independent set of written responses to different open-ended questions. We found that the classifier achieved an F1-score = 79.15% for incoherent detection, which is better than baselines using different heuristics.

**Keywords:** written short answers; incoherent answer detection; natural language processing; arguing and communication; elementary school; mathematics education; online platform; deep learning; open-ended questions

## 1. Introduction

Arguing and communicating are basic skills in the mathematics curriculum. For example, in the U.S. Common Core State Standards for Mathematics (CCSSM) [1], it is stated that students should, "construct viable arguments and critique the reasoning of others". According to the CCSSM, mathematically proficient students should "understand and use stated assumptions, definitions, and previously established results in constructing arguments". In the case of elementary students, they should be able to "construct arguments using concrete referents such as objects, drawings, diagrams, and actions. Such arguments can make sense and be correct, even though they are not generalized or made formal until later grades". The CCSSM establishes that all students at all grades should be able to "listen or read the arguments of others, decide whether they make sense, and ask useful questions to clarify or improve the arguments". In Chile, the National Standards for Mathematics also has arguing and communication as one of the four core mathematics skills for all grades. An extensive literature supports the inclusion of these elements in the mathematics curricula of different countries, emphasizing the importance of developing the ability to argue and communicate in mathematics. For example, [2] states that students in grades 3 to 5 should learn to create general arguments and learn to critique their own and others' reasoning. Mathematics instruction should help students learn to communicate solutions with teachers and with peers [3,4].

However, this is not an easy task. For example, an analysis of third-grade German textbooks found that no more than 5–10% of all textbook tasks ask for reasoning [5]. The same situation happens in other countries. In Chile, for example, textbooks and national standardized tests do not have explicit reasoning or communication questions.

On the other hand, the process of arguing and communicating in writing has several additional advantages over doing so only verbally. It allows students to reason immediately and visually about the correctness of their solution [6]. It also supports reasoning and the building of extended chains of arguments. Writing facilitates critique of the reasoning of others [1], reviewing the argumentation of peers, and receiving feedback from them. Although writing in mathematics can serve many purposes [7], in this paper, we focus primarily on explaining how the student arrived at the solution.

Developing arguing and communication competencies in mathematics for elementary school classrooms is a great challenge. If, in addition, the teacher wants the students to do so in writing, then there are several additional implementation challenges. It requires at least two conditions: all students should be able to write their answers and all should be able to comment on answers written by peers. At the same time, they should receive immediate feedback.

One solution is to use online platforms. The teacher poses a question and in real time receives the answers from the students. Giving feedback should take one or two minutes. This is possible since fourth graders in our population typically write answers of eight to nine words. However, reviewing the answers in their notebook or smartphone is very demanding for the teacher. The teacher must review 30 answers in real time. To facilitate the revision, the first task is to automate the detection of incoherent answers. These answers can reflect a negative attitude of the student. They can also show an intention not to respond. Thus, automatic detection enables the teacher to immediately require correction of them.

Some incoherent answers are obvious. For example, "jajajajah". Others are more complex. They require some degree of understanding of the question and the answer and the ability to compare them.

In this work, we use the ConectaIdeas online platform [8]. ConectaIdeas is an online platform where students answer closed and open-ended questions [9]. On ConectaIdeas, teachers develop their own questions, designing them from scratch or they select them from a library of questions designed previously by other teachers. Then the teachers use these questions to build their own formative assessments for 90 min sessions. The formative assessments comprise 20 to 30 closed questions and one or two open-ended questions. Students answer them in laboratory sessions held once or twice a week, or at home. The teacher checks that the open responses are coherent. If there are some answers that are not coherent, then the teacher asks the student to answer again. Once all the answers are coherent, then the teacher assigns them randomly for peer review.

The answers of fourth-graders from at-risk schools are very short. The average number of words in the responses is eight to nine words. This average increases as the school year progresses [10,11]. On the other hand, in previous randomized controlled trials (RCT) using the ConectaIdeas platform with fourth graders, we have found that the length of the responses to open-ended questions has a significant and positive effect on end-of-year learning in math [9].

To analyze students' answers it is critical to be able to classify questions. The accuracy of question classification plays a key role in detecting if the response is coherent or not. There is a long history of the study and classification of questions. Already in 1912 [12], stressed that "the subject of questioning should have a place in the training of every teacher—a place that is comparable in importance with 'fund of knowledge' and psychology" p. 5. The authors of [12] categorized teachers' questions, producing statistics for six different subjects, including middle and high school science and mathematics. In the aforementioned study, the authors counted the number of questions asked per session and the number of questions related to memory. The study also considered the quality of the questions.

In 1970, ref. [13] concluded that in the previous half-century, there had been no essential change in the types of questions that teachers emphasized in the classroom.

Thus, as part of the development of an answer classifier, it is necessary to develop an automatic classifier for open-ended questions in fourth-grade mathematics.

In this paper, we analyzed 716 open-ended questions and their corresponding 14,457 written answers. A total of 974 fourth graders wrote these answers on the ConectaIdeas online platform. We found that 13% of the answers were incoherent.

A unique characteristic of this paper is that the research is based on open-ended questions designed and written by teachers on the spot. Therefore, there are two basic features that make it more challenging to build automatic classifiers. First, unlike other studies [14], the questions are not reviewed and validated by third parties. Some may be poorly written or even contain errors. Second, there are very few answers per question. Typically, the number of responses corresponds to the answers of 20 to 60 students, belonging to one or two courses on which the teacher teaches. These two characteristics are typical of those found in almost all classes in real-world classrooms. The low number of responses per question makes it difficult to use deep learning algorithms. The difficulty is that these algorithms require a large number of responses per question in order to identify statistically valid patterns.

Our research question was: To what extent can we construct an automatic classifier that can detect, in real-time, incoherent answers given by fourth graders to open-ended math questions created and verbalized on the spot by the teacher on an online platform?

The plan was first to build a classifier of open-ended fourth-grade mathematics questions. Second, in a manual review, a researcher from the team identified six types of questions. Third, we manually labeled 716 questions. Fourth, we manually classified the 14,457 responses obtained as coherent or non-coherent. Some responses were clearly incoherent regardless of the question. Fifth, we built a classifier for obviously incoherent responses. Sixth, we built a classifier of coherence or non-coherence answers for each type of question. Finally, we built a wide range of classifiers. We tested each classifier on an independent set of questions and their corresponding answers.

## 2. Related Works

First, we review the classification of questions, and then we review the classification of answers. We are not aware of the development of automatic classifiers of mathematical questions formulated and asked by elementary school teachers. However, there are several developments of question classifiers in other applications. The automatic classification of questions is an area of natural language processing (NLP) that has developed greatly in recent times. We review some question clarifiers as they illustrate the main algorithms used. Question classifier systems may improve the performance of question-answering systems. However, none of these systems focuses on detecting incoherent answers.

Question type classification is a particular form of text classification within NLP [15]. In our case, we consider six types of questions in order to detect what type a new question is. We use only information from the question itself. The literature on question classification considers various techniques, from machine learning (ML) with bags-of-words [16] to deep learning [17].

Ref. [18] reports the building of a question classifier for six types of questions: Yes-No questions (confirmation questions), Wh-questions (factoid questions), choice questions, hypothetical questions, causal questions, and list questions, for 12 types of term categories, such as health, sports, arts, and entertainment. To classify question types, the authors use sentence representation based on grammatical attributes. Using domain-specific types of common nouns, numeral numbers, and proper nouns and ML algorithms, they produced a classifier with 90.1% accuracy.

Ref. [19] combined lexical, syntactic, and semantic features to build question classifiers. They classified questions into six broad classes of questions, and each of these into several more refined types of questions. The authors applied nearest neighbors (NN), naïve Bayes

(NB), and support vector machine (SVM) algorithms, using bag-of-words and bag-of-n-grams. They obtained 96.2% and 91.1% accuracy for coarse- and fine-grained question classification.

For linguistic features extraction, we consider part-of-speech (POS) and dependency tags for the question using the Spanish version of the Spacy library (authors on https://explosion.ai/, accessed on 16 March 2022).

Some authors use the BERT model to represent questions as vectors and create classifiers. These representations have obtained outstanding results in text classification when compared to traditional machine learning [20]. Ref. [21] reported the development of a BERT-based classifier for agricultural questions relating to the Common Crop Disease Question Dataset (CCDQD).

A very high accuracy of 92.46%, a precision of 92.59%, a recall of 91.26%, and a weighted harmonic mean of accuracy and recall of 91.92% were obtained. The authors found that the BERT-based fine-tuning classifier had a simpler structure, fewer parameters, and a higher speed than the other two classifiers tested on the CCDQD database: the bidirectional long short-term memory (Bi-LSTM) self-attention network classification model and the Transformer classification model.

Ref. [22] reported the use of a Swedish database with 5500 training questions and 500 test questions. The taxonomy was hierarchical with six coarse-grained classes of questions: location, human, description, entity, abbreviation, and number. It also included 50 fine-grained classes of questions. Two BERT-based classifiers were built. Both classifiers outperformed human classification.

Ref. [23] reported the building of an SVM model to classify questions into 11 classes: advantage/disadvantage, cause and effect, comparison, definition, example, explanation, identification, list, opinion, rationale, and significance. The authors tested the classifiers on a sample of 1000 open-ended questions that they either created or obtained from various textbooks. For our work, one relevant class of question is an explanation. They are common in our database. To classify explanations, they obtained an accuracy of 83.3%, a precision of 83.3%, and a recall of 50.0%.

In answer classification, there have been several reported studies, although not for answer coherence.

The authors of ref. [24] used NLP algorithms to assess language production in e-mail messages sent by elementary students on an online tutoring system during the course of a year. They found that lexical and syntactic features were significant predictors of math success. In their work, the students did not answer questions. Therefore, it was not possible to verify if the answers were coherent.

The authors of ref. [25] analyzed 477 written justifications of 243 third, fourth, and sixth graders, and found that these could be accounted for by a one-dimensional construct with regard to a model of reasoning. However, they did not code incoherent responses. The absence of incoherent answers may be due to the nature of this project. In a small research project, the behavior of the students is different. In their handwritten answers, the students wrote only coherent answers. In contrast, in a large-scale project where students write every week, students may behave differently.

Ref. [26] explored the impact of misspelled words (MSW) on automated computer scoring systems in the context of scientific explanations. The results showed that, while English language learners (ELLs) produced twice as many MSW as non-ELLs, MSW was relatively uncommon in the corpora. They found that MSW in the corpora is an important feature of computer scoring models. Linguistic and concept redundancy in student responses explained the weak connection between MSW and scoring accuracy. This study focused on the impact of poorly written responses but did not examine answers that may have been incoherent or irrelevant to the open-ended questions.

There is an extensive literature on automated short answer grading (ASAG). In ref. [27], it was found that automated scoring systems with simple hand-feature extraction were able to accurately assess the coherence of written responses to open-ended questions. The

study revealed that a training sample of 800 or more human-scored student responses per question was necessary to accurately construct scoring models, and that there was nearly perfect agreement between human and computer-automated scoring based on both holistic and analytic scores. These results indicate that automated scoring systems can provide feedback to students and guide science instruction on argumentation.

The authors of ref. [28] identified two main challenges intrinsic to the ASAG task: (1) students may express the same concept or intent through different words, sentence structures, and grammatical orders, and (2) it can be difficult to distinguish between nonsense and relevant answers, as well as attempts to fool the system. Existing methods may not be able to account for these problem cases, highlighting the importance of including such considerations in automated scoring systems.

Finally, NLP has had a significant impact on text classification in educational data mining, particularly in the context of question and answer classification [28]. Shallow models, such as machine learning algorithms using bags-of-words and bag-of-n-grams, have been employed to classify question types with high accuracy. Deep learning models, including BERT-based classifiers [29,30], have shown outstanding performance in representing and classifying questions, outperforming traditional machine learning approaches [28]. Ensemble models, such as XGBoost classifiers [31], have also been utilized for question classification, achieving impressive results [32]. In answer classification, studies have explored the use of NLP algorithms to assess language production and reasoning in student responses. However, few studies have specifically focused on detecting incoherent answers. The research conducted in this study fills this gap by utilizing NLP and machine learning algorithms to build automatic classifiers for detecting incoherent written math answers, contributing to the field of mathematics education and providing valuable insights for educational assessment systems.

## 3. Materials and Methods

We used questions and answers in the ConectaIdeas online platform. ConectaIdeas has a series of mathematics exercises for elementary school students. Once or twice a week, each student answered questions during each session, which typically lasted 90 min. The questions were either closed-ended (e.g., multiple-choice questions) or open-ended (e.g., essay questions). Open-ended questions received responses in unstructured written form. We used NLP techniques to extract useful information from written responses. NLP detects topics covered, and specific words that help evaluate the quality of the responses.

We used a combination of traditional and recent techniques to detect incoherence in answers to open-ended questions. Figure 1 illustrates the approach. It involved collecting question-answer pairs, labeling them, applying several types of ML and NLP algorithms to build classifiers, and then assessing the performance of the different classifiers.

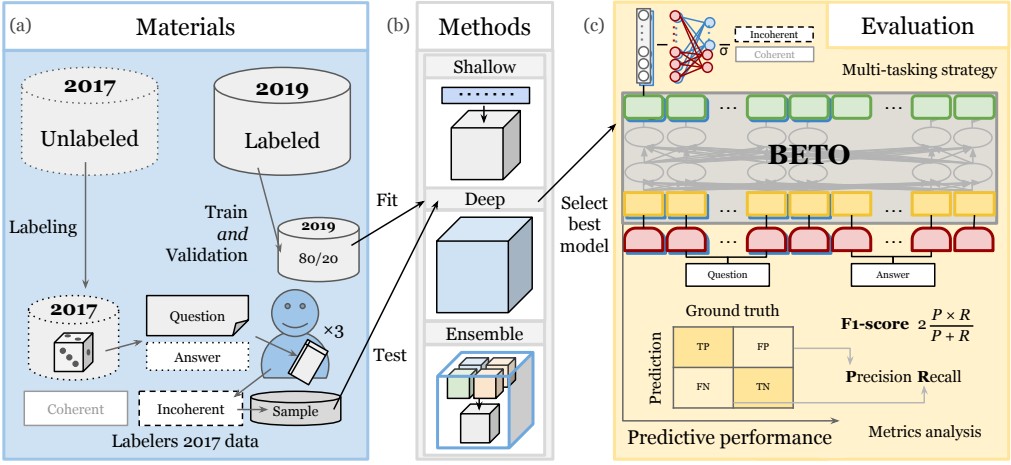

**Figure 1.** (**a**) Datasets and labeling. (**b**) Three types of ML algorithms. (**c**) Performance evaluation.



### 3.1. Materials

The dataset contained questions and answers from 2017 and 2019. Teachers created one open question on the spot in each session. Some questions were conceptual while others presented situations and characters. Additionally, some questions explicitly asked for explanations. We identified six types of questions (Figure 2).

Questions of type Q1 (Calculate without explaining) require the calculation of a quantity without an explanation or justification of the answer. Questions of type Q2 (Calculate with explaining) ask for the calculation of a quantity, but also require writing of an explanation or justification of the answer. Questions of type Q3 (Choice and/or affirmation) involve introducing characters and statements. This can take the form of deciding who is right or whether a character's statement is correct. In both cases, it asks for a justification for the answer. Questions of type Q4 (Compare quantities) ask to compare two quantities and name which or why one is greater than the other one. Questions of type Q5 (Procedure and content knowledge) ask to write a problem or ask content questions. Questions of type Q0 (Others) are questions that do not fall into any of the previous categories.

The linguistic features of the responses depended on the nature of the question. In some answers, students had to write integers, decimals, or fractions. In some answers, they had to write explanations. In others, they had to name one or more characters. In some answers, they had to write a keyword, such as "yes" or "no".

We defined two types of incoherent answers (Figure 2): Question-independent incoherent answers. For example, answers with faces, laughter, curse words, phonemic errors, omitting letters, transposing consonants, and pasting words; Question-dependent incoherent answers. In this case, the question is necessary. For example, "no" may be considered incoherent for type Q1 questions.

### 3.2. Dataset Description

In the first stage, for the 2019 data, three professors and a member of the research team worked on labeling pairs of questions and answers. They labeled the type of answer and whether the answer was coherent or not. In the second stage, also for the 2019 data, only one member of the team participated and not the teachers. This was because there was approximately a year between the first and second stages. At this stage, we labeled whether the answer was incoherent independent of having access to the question or not. Then, in the third stage, six new teachers participated in labeling the data for the 2017 pairs of questions and answers. The new teachers were trained based on examples from the first stage to detect coherence/incoherence.

For 2019, we labeled 14,457 answers according to the two types of coherence, and labeled 716 questions according to the six question types. For 2017, we collected 1180 questions and 16,618 answers.

Of the 716 questions in 2019, we labeled 30.16% as type Q2 (Calculate with explaining) and 29.18% as type Q3 (Choice and/or affirmation). Type Q0 (Others) questions were the least common (Tables 1 and 2). On the other hand, of the 14,457 responses from 2019, we identified 13.33% as incoherent, of which 77.17% were question-dependent.

Since most of the responses were coherent, the dataset was highly unbalanced. It is therefore necessary to consider this imbalance in the construction of classifiers.

Once we built the different classifiers based on the 2019 data, we tested them on the 2017 data. We randomly sampled 3000 answers from the 16,618 responses of 2017. Six teachers labeled the responses according to a rubric. We trained the teachers with 44 pairs of questions and answers. Then we held a discussion session to address any questions they had about the criteria and examples.

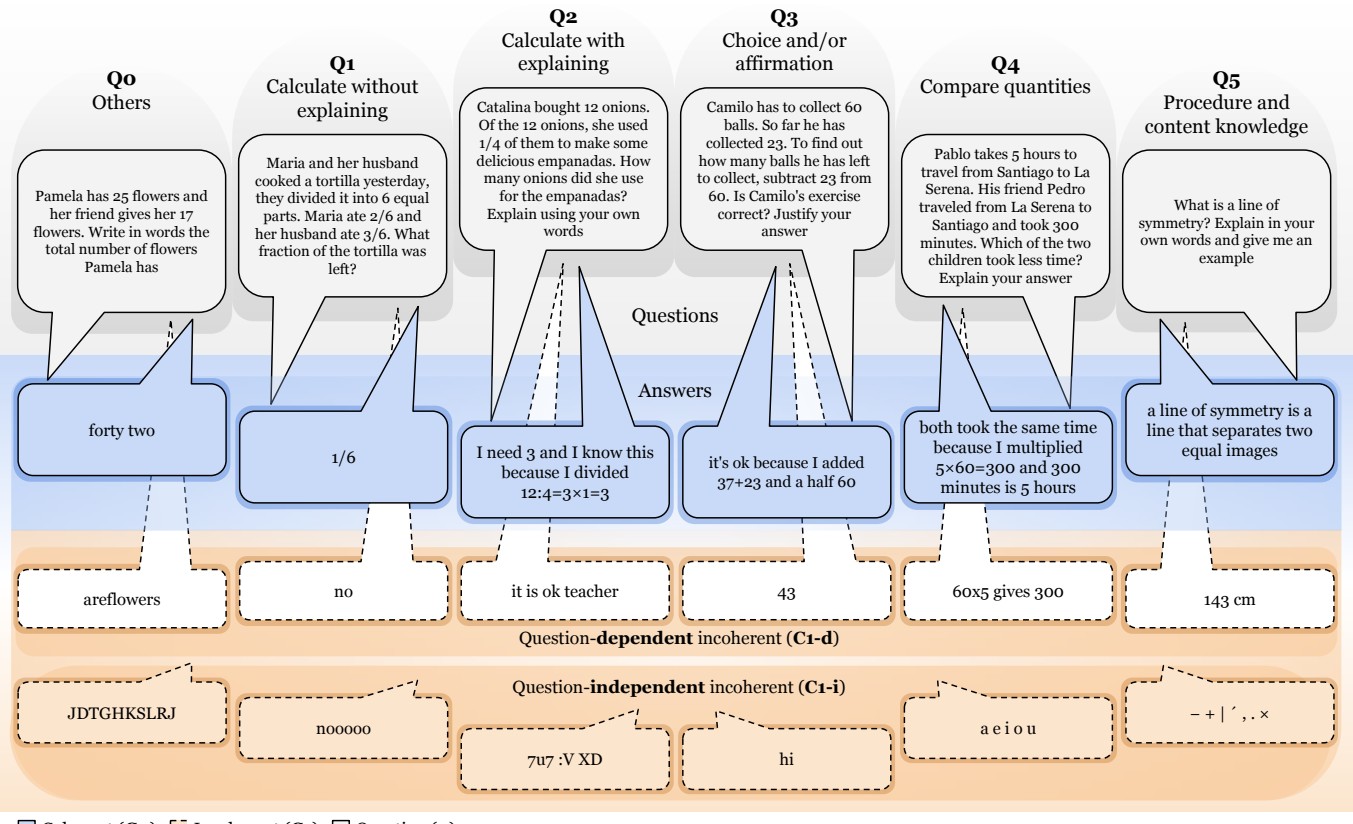

**Figure 2.** Example of open-ended math exercise questions and fourth graders' answers. The first row contains the questions, separated by type and indexed with $Q_i$, where *i* is a number between 0 and 5. The second row has coherent answers, and the third row has incoherent answers dependent on the question. The fourth row contains incoherent answers that are independent of the question. Note: Examples originally in Spanish.

Teachers selected the order of the pairs to label. Three of the teachers labeled more than 1000 responses, while the other three labeled less than 800. However, when looking at the incoherence percentages, the three teachers who labeled the most had similar incoherence rates, whereas one of the three teachers who labeled the least had a considerably higher proportion of incoherent answers. This was due to some of the labeler's strategies of first tagging the responses that were clearly identified as incoherent.

The three teachers agreed on 62.8% (677/1078) of the 1078 common responses they labeled. We used these 677 responses with the same label as the ground truth to evaluate the classifier. The teachers judged that 20.0% (136/677) of them were incoherent (Tables 1 and 2). This percentage was higher than the 13.3% incoherence percentage obtained for the 2019 data.

**Table 1.** Number of answers (with percentages), and below the proportion of coherence, per type of question and year.

| Year | Q0 | Q1 | Q2 | Q3 | Q4 | Q5 |
|------|------|------|------|------|------|------|
| 2019 | 453 (3.13%) 87.85% | 1939 (13.41%) 95.20% | 4812 (33.28%) 84.51% | 4061 (28.09%) 83.84% | 1638 (11.33%) 84.98% | 1554 (10.74%) 91.44% |
| 2017 | 149 (22.00%) 71.81% | 9 (1.32%) 88.88% | 57 (8.41%) 68.42% | 303 (44.75%) 86.46% | 16 (2.36%) 68.75% | 143 (21.12%) 79.72% |

**Table 2.** Number of questions (with percentages) per type of question and year.

| Year | Q0 | Q1 | Q2 | Q3 | Q4 | Q5 |
|------|-----|-----|-----|-----|-----|-----|
| 2019 | 27 (3.77%) | 114 (15.92%) | 216 (30.16%) | 209 (29.18%) | 70 (9.77%) | 80 (11.17%) |
| 2017 | 68 (18.88%) | 5 (1.38%) | 35 (9.72%) | 161 (44.72%) | 9 (2.5%) | 82 (22.77%) |

Table 1 contains the distribution of the types of questions for the years 2019 and 2017. Specifically, the table shows in the first row the number of responses, in the second row it shows the percentage of responses for each type of question, and in the third row shows the proportion of consistent responses for each question type.

In 2019, the authorities asked teachers to produce more complex questions, that is questions that require explaining how the student arrived at the result. For this reason, in 2019, there were few Q1 answers (Calculate without explaining). On the other hand, teachers were also asked to create more complex questions, like Q3, and to stop asking simple questions, like Q4.

In 2017, the table represents all questions and their answers, while in 2019, the table represents a random sample.

*3.3. Tasks*

The detection of incoherent answers was the main task. However, we also needed to classify the type of each question.

We represented open-ended questions by $q$, with a type $Q$ ranging from 0 to 5. We represented an answer to an open-ended question by $a$. If an answer was coherent for a given question, we denoted it as $(q, a) \in C0$. If it was incoherent, we denoted it as $(q, a) \in C1$. Incoherent answers were further divided into question-dependent incoherence, denoted $(q, a) \in C1 - d$, and question-independent incoherence, denoted $(q, a) \in C1 - i$.

Using this notation, we defined four classifier functions, denoted by $\delta_{C1}$, $\delta_{QT}$, $\delta_{C1-i}$, and $\delta_{C1-d|q \in Q}$. The first one ($\delta_{C1}$) was our main task, and the other ones ($\delta_{QT}$, $\delta_{C1-i}$, and $\delta_{C1-d|q \in Q}$) were the secondary tasks. The incoherence classifier function $\delta_{C1}$ receives a $(q, a)$ question-answer pair and indicates with a 1 whether the answer $a$ is incoherent for the question $q$ and 0 otherwise. The question type classifier function $\delta_{QT}$ receives a question q and returns a natural number between 0 and 5 indicating one of six possible question types. The question-independent incoherence classifier function $\delta_{C1-i}$ receives a $(q, a)$ pair and indicates with 1 if the answer a is incoherent independent of the question $q$. Finally, the question-dependent incoherence classifier function $\delta_{C1-d|q \in Q}$ indicates with a 1 whether an answer is incoherent for a given question type $Q$.

*3.4. Proposed Methods*

We considered three categories of models to build the classifiers: shallow models, deep models, and ensemble models (Figure 3).

First, we built shallow models (Figure 4). Shallow model methods for NLP tasks are methods that combine a text representation created through expert-guided feature extraction and shallow classifiers that learn to discriminate using this representation of the data. Examples of feature-extraction techniques include TF-IDF [33], word embeddings [28,34], and POS tagging [35]. Examples of shallow classifiers include SVMs [36,37] and NB [38]. In our study, for feature extraction, we experimented with both hand-engineered features and representations learned from word embeddings of the BETO model [39]. BETO is the Spanish version of BERT. It uses bidirectional encoder representations from transformers [40]. Additionally, we considered the combination of both types of representations. We examined two classifiers: a kernel-based algorithm called SVM and a tree-based model called XGBoost [31].

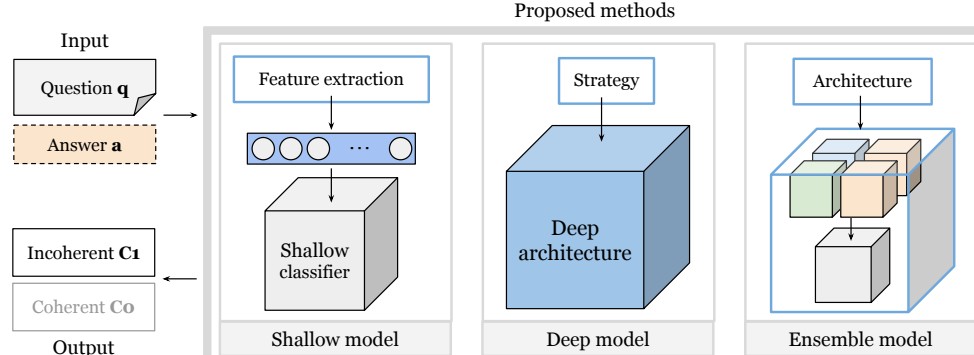

**Figure 3.** Shallow models are low complexity and suitable for simple tasks with limited data, capturing only shallow patterns. Deep models are more complex, capturing more intricate patterns. Ensemble models use a combination of shallow and deep models, depending on the task.

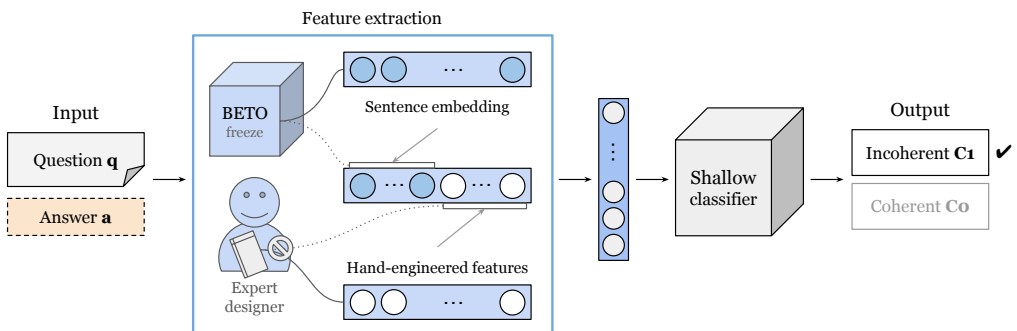

**Figure 4.** Classifying incoherence in open-ended answers using various feature extraction methods, including handcrafted features and word embeddings. Interpretable features, such as word count, question-answer overlap, and linguistic knowledge, were used to identify incoherence. The study utilized the Spanish version of BERT (BETO) for vector representations of text (Figure 5) and two classification models, support vector machines (SVMs) and eXtreme Gradient Boosting (XGBoost). We evaluated the models and fit the parameters following Tables 3 and 4. Hand-engineered features are described in Table 5.

**Table 3.** Outline of the parameter grid for an SVM model [36,41]. It contains four parameters—kernel, class-weight, gamma, and C—each with a set of possible values. The kernel parameter defines the type of kernel used. The class-weight parameter assigns different weights to different classes. The gamma parameter defines the width of the kernel function. The C parameter controls the misclassification penalty.

| Kernel | Class-Weight | Gamma | C |
| --- | --- | --- | --- |
| `{linear, rbf}` | `{balanced, None}` | `{scale, auto}` | {1, 10, 15} |

**Table 4.** Outline of the parameter grid for the XGBoost model [31]. The table includes three experiments with parameters max-depth, n-estimators, and learning-rate. Max-depth controls the maximum depth of each tree, n-estimators is the number of trees, and learning-rate controls the step size used to update the weights.

| Experiment | Max-Depth | n-Estimators | Learning-Rate |
| --- | --- | --- | --- |
| XGBoost + HF | {2, 4, 6, 8, 10} | {100, 75, 50, 25, 10} | {0.001, 0.03, 0.1} |
| XGBoost + Mix | {6, 8, 10} | {125, 100, 75} | {0.03, 0.1, 0.3} |
| General (XGBoost) | {6, 8, 10} | {150, 125, 100} | {0.1, 0.3, 0.5} |

**Table 5.** Description of type of attributes based on hand-engineered features.

| Type | Description |
| --- | --- |
| Answer-level | Use information from the entire answer. We analyze the length, punctuation, alphabetic and numerical attributes. Length is measured in terms of characters or tokens and indicates if an answer is too short or long. Punctuation use is checked for the correctness of mathematical symbols and the presence of rare ones. Alphabetic attributes detect non-words or spelling mistakes, while numerical ones check for appropriate numbers and their spelling accuracy. Misspelled numerical representations are corrected and transformed into numbers using a triangular matrix algorithm [42] (available on https://github.com/furrutiav/NumberSpellChecker). |
| Token-level | Only use information from individual tokens in the answers. These include character repetition and character frequency. Character repetition looks at the consecutive repetition of letters in a token. It can indicate if a response contains non-words or exaggerated exclamations. Character frequency looks at the frequency of letters in a token and can help to identify if a word contains rare letters that would not normally be found within a Spanish word. |
| Semantic | Consider the meaning of some tokens and phrases. Some tokens represent faces and others do not belong to official dictionaries. We detail each of these below: We use dictionaries like the Real Academia Española (available in https://pypi.org/project/pyrae/) and the UrbanDictionary (available in http://api.urbandictionary.com/v0). They help detect words that are spelled correctly, colloquial tokens, and slang. Emoticons and emojis [43] help identify incoherent responses. Keywords, such as curse words, also help identify incoherent responses. |
| Contextual | Detecting coherent and incoherent written responses can be achieved by analyzing certain attributes, such as binary words, key questions, and overlap between answer and question tokens. Some questions have certain types of tokens appearing in the answer, like character names. Key questions require specific keywords in the question, while overlap measures the similarity between questions and answers. Proper nouns and nominal subjects in the question can also aid in detecting certain types of questions. These attributes can be computed using methods such as counting common tokens or using fuzzy sets and the Levenshtein distance [44,45]. |
| Linguistic | Capture useful information from the answers. We use the Spanish version of the Spacy library to extract the following linguistic attributes: shape, alpha, stop-words. Part-of-speech (POS) and dependency tags are important attributes for detecting coherent and incoherent written responses. Shape refers to the shape of the tokens. Stop-words are words that are noted for their abundance. We used POS tags to detect each token in a response (available in https://universaldependencies.org/u/pos/). Dependency tags are syntactic dependencies between the tokens in the answer (available in https://universaldependencies.org/u/dep/). |

Note: All links were accessed on 16 March 2022.

Second, we built deep models. Deep learning has been extremely successful in the area of NLP, for example, in language models based on transformer architectures. Unlike other deep learning models with neural networks in text, such as convolutional and recurrent, transformers use the attention mechanism [29]. Among the models that have demonstrated outstanding performance in a number of NLP tasks and that use the attention mechanism

is the BERT language model [30]. These models learn the appropriate representations of the data rather than using representations preconceived by an expert. In particular, we use the BETO language model (Figure 5). We used two strategies to train the BETO model: fine-tuning [39,46] and multi-tasking [47,48]. Fine-tuning involves adjusting the model to the main task by adding a last classifying layer. Multi-tasking involves considering a secondary task (question type prediction) as a regularizer for the main task.

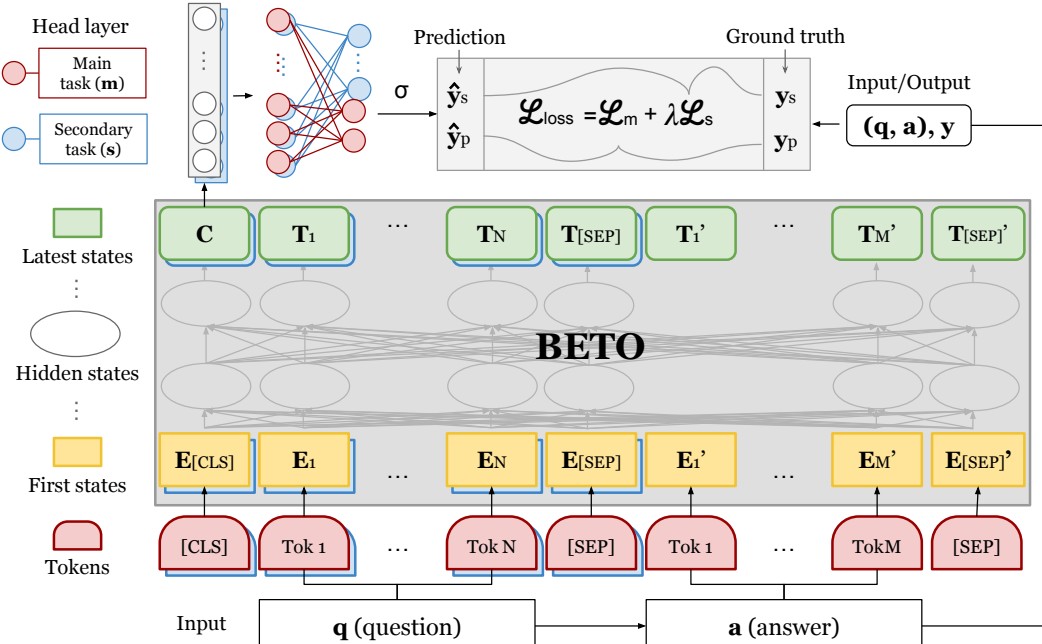

**Figure 5.** The BETO model was trained to classify incoherence in open-ended question answers using multi-tasking and fine-tuning strategies with a fixed $\lambda = 0$. The model was pretrained on two self-supervised tasks, the masked language model and next sentence prediction, using the sources of the OPUS Project and Spanish Wikipedia datasets. Fine-tuning involved adjusting the pretrained model by adding a last linear layer, while multi-tasking trained the model on multiple tasks simultaneously. Further pretraining involved retraining the model with intrinsic tasks specific to the domain of the main task data. For the BETO multi-tasking experiment, a BETO model was trained directly without further pretraining [46], using specific parameters (Table 6).

**Table 6.** Outline of the parameter grid for the BETO model. It uses a batch size of 64, a cross-entropy criterion, and a learning rate of $2 \times 10^{-5}$ for training. For fine-tuning, the number of epochs is set to 4, and for multi-tasking, the number of epochs is set to 6. The weights are balanced at $1/6.5$, and the input length is set to 60. To increase training speed, we trained the model on a Graphics Processing Unit (GPU) with a dropout rate of 0.1. We used the warm-up proportion of 0.1 in order to reduce overfitting [40]. Note: (ft) fine-tuning and (mt) multi-tasking

| Batch Size | Criterion | Epochs | Learning Rate | Weights | Length |
|------------|-----------|--------|---------------|---------|--------|
| 64 | Cross Entropy | 4 (ft), 6 (mt) | $2 \times 10^{-5}$ | balanced $1/6.5$ | 60 |

Third, we used ensemble models. Ensemble models are a powerful tool for improving the predictive performance of a single model by combining the predictions of multiple models [32]. These methods have been widely used to address a variety of challenges in machine learning, such as improving accuracy, reducing bias, and providing an overall better model. Ensemble models typically involve training multiple models. Each model can have slightly different configurations. Combining their predictions, they create a more robust and accurate result than any single model could produce on its own. In our study, we designed two different ensemble models. The first was a logical architecture based

on simple and intuitive decisions (Figure 6). The second was a general architecture that uses the same individual classifiers as the logical architecture (Figure 7). However, the architectures were different because we assembled the latter by training an XGBoost model.

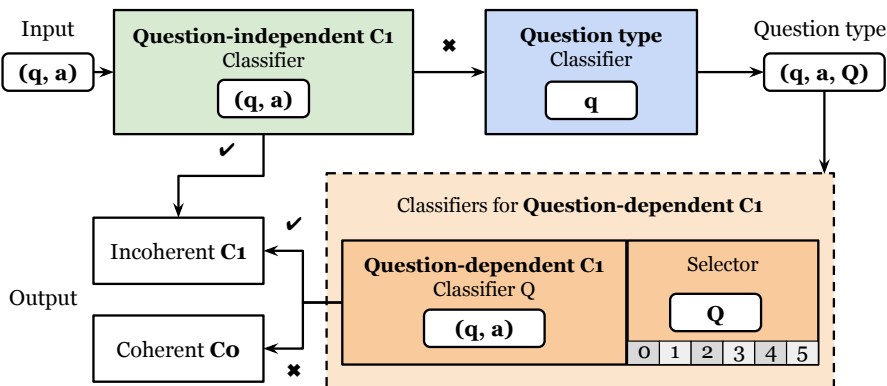

**Figure 6.** The logical architecture is an ensemble model consisting of eight classifiers designed to identify incoherent answers to different types of questions. Question-dependent incoherence (C1-d) requires further analysis than question-independent incoherence (C1-i). The model includes a C1-i classifier, a question type classifier (QT), and a C1-d classifier per type of question. The QT classifier is a BETO model trained with a fine-tuning strategy, while the C1-i and C1-d classifiers are XGBoost models using hand-crafted features and BETO sentence embeddings. By considering the type of question, the logical architecture can determine whether an answer is coherent or not.

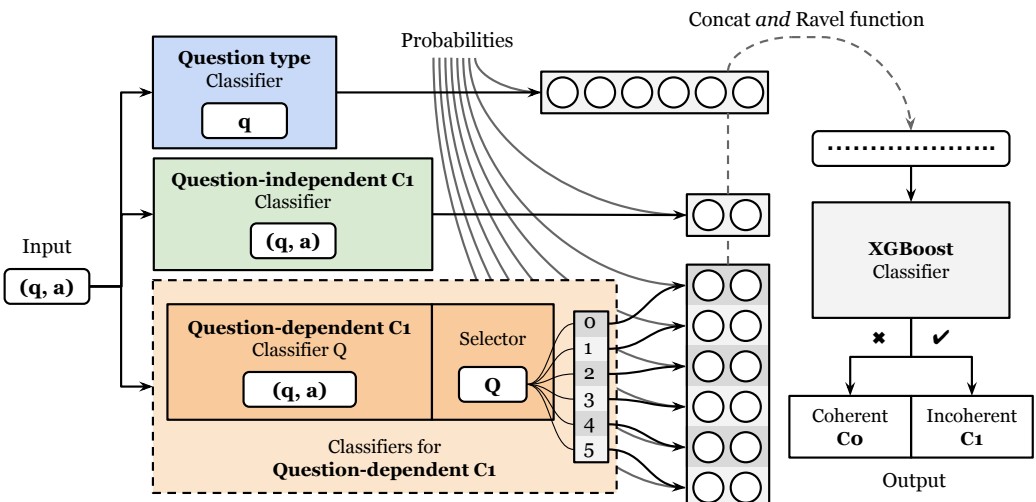

**Figure 7.** The general architecture incorporates the same classifiers as the logical architecture but requires additional training to properly integrate them. This approach uses the probabilities associated with each classifier to generate a large probability vector, which is then inputted into a tree-based model to make the best decision and determine whether the answer is coherent with the question. The general architecture is an ensemble model that includes the same eight classifiers as the logical architecture (Figure 6). To optimize the XGBoost model's parameters (Table 4), we use a grid search method over the validation set to choose the best generalizing model.

*3.5. Evaluation*

We split the dataset into training, validation, and test sets. Then we compared the model's performance against baseline models on these datasets. Given the significant imbalance of the incoherent answers (13% of the answers), the percentage of correct predictions (accuracy) is an inappropriate metric. Instead, we used three performance metrics: precision, recall, and the F1-score.

Precision ($P$), the proportion of answers correctly predicted as incoherent between all the answers predicted as incoherent, $P = TP/(TP + FP)$, where $TP$ is true positives and $FP$ is false positives. Recall ($R$), the proportion of answers correctly predicted as incoherent between all the incoherent answers, $R = TP/(TP + FN)$, where $FN$ is false negatives. F1-score ($F$), the harmonic mean between precision and recall: $F = 2(P \times R)/(P + R)$.

We also kept track of the support ($S$), the number of occurrences of each particular class in the true responses.

We used the 2019 data for training and validation, while we used the 2017 subsample for testing. The 2017 data allowed us to test the model in a real-world scenario. Not only was it a different year. It was also a different set of students.

Additionally, we stratified the 2019 data by question. This means no two answers to the same question were in different training and validation sets. We split into training and validation and by question type, with 80% of the questions and their answers chosen as a training part and 20% as a validation part. The training set contained 10,019 coherent and 1540 incoherent question-answer pairs; the validation set had 2510 coherent and 388 incoherent pairs; and the test set contained 541 coherent and 136 incoherent pairs.

## 4. Results

We present the performance of the baseline, shallow, deep, and ensemble models. We have included the information on data preprocessing, model implementations, training code, model predictions, and results, in the GitHub repository https://github.com/furrutiav/coherence-prediction-systems-2023 (accessed on 17 April 2023). Additionally, to implement the shallow models, determine the performance metrics, and search for parameters, we relied on Scikit-learn [49]. For the management of neural models, we relied on Huggingface Transformers [50] and PyTorch [51]. Table 7 shows the models' classification performances of answers according to incoherence (positive class) and their metrics (precision, recall, and F1-score) in the three-set scheme (test/validation/train).

**Table 7.** Predictive performance of the models on the test set, validation set, and training set. The category column corresponds to baselines and the three families of proposed models: shallow, deep, and ensemble models. The model column refers to the names of the experiment models, and the numerical values are the metrics associated with each dataset and model.

| Category | Model | Test Set | | | Validation Set | | | Train Set | | |
|---|---|---|---|---|---|---|---|---|---|---|
| | | *P* | *R* | *F* | *P* | *R* | *F* | *P* | *R* | *F* |
| Baseline | Dummy (most frequent) | 0.00 | 0.00 | 0.00 | 0.00 | 0.00 | 0.00 | 0.00 | 0.00 | 0.00 |
| | Dummy (stratified) | 20.24 | 12.50 | 15.45 | 15.06 | 15.72 | 15.38 | 13.48 | 13.57 | 13.53 |
| | Dummy (uniform) | 21.65 | 52.21 | 30.60 | 13.57 | 49.48 | 21.30 | 13.00 | 48.90 | 20.54 |
| | NB + BOW | 100.00 | 2.21 | 4.32 | 60.99 | 28.61 | 38.95 | 80.06 | 55.00 | **65.20** |
| | **Rule-based** | 69.79 | 49.26 | **57.76** | 50.46 | 56.96 | **53.51** | 48.07 | 53.38 | 50.58 |
| Shallow | XGBoost + HF | 70.00 | 72.06 | 71.01 | 89.63 | 80.15 | 84.63 | 94.80 | 92.40 | 93.59 |
| | **XGBoost + Mix** | 70.92 | 73.53 | **72.20** | 93.97 | 84.28 | **88.86** | 99.93 | 99.74 | **99.84** |
| | SVM + BETO embeddings | 60.78 | 68.38 | 64.36 | 88.05 | 72.16 | 79.32 | 93.27 | 80.13 | 86.20 |
| **Deep** | BETO fine-tuning | 74.83 | 78.68 | 76.70 | 81.23 | 84.79 | 82.98 | 98.21 | 99.94 | **99.07** |
| | **BETO multi-tasking** | 76.19 | 82.35 | 79.15 * | 84.50 | 84.28 | **84.39** | 96.25 | 100.00 | 98.09 |
| Ensemble | Logical | 66.45 | 75.74 | 70.79 | 88.25 | 90.98 | 89.59 | 92.67 | 98.51 | 95.50 |
| | **General (XGBoost)** | 78.79 | 76.47 | **77.61** | 94.49 | 92.78 | **93.63** | 99.93 | 99.74 | **99.84** |

Note: Precision ($P$), Recall ($R$), F1-score ($F$); Support: 136 (test), 388 (val), 1540 (train). The **bold** category designates the category exhibiting the highest-performing model among all categories. Likewise, the bold models represent the models within each category that demonstrate the best performance. In contrast, the bold F1-score values indicate the highest score within each category. Additionally, the single value marked with an asterisk (*) corresponds to the highest F1-score value across all categories in the test set.

### 4.1. Baseline Models

We consider three dummy models and two strong models for our baseline. The first three are the most frequent classifier, the uniform classifier that with a 50% probability randomly assigns coherent or incoherent labels, and the stratified classifier, which randomly

assigns coherent or incoherent labels with a probability proportional to their rate on the training dataset (see Appendix A.1). They all predict the class independently of the question-answer pair.

The other two models are a multinomial NB and a rule-based model. The NB model is a supervised learning method for text classification, where, given a text x, it seeks to find the class c that maximizes the probability of the class given the input [52]. The rule-based model is a stronger base model, which uses an if/else structure and both the question and the answer to detect coherence. We specify these models in the Appendix A.1.

The most frequent class model has zero performance. This is because it only predicts the majority class and never gets true positives ($TP$). The stratified model performs worse than the uniform model due to its lower rate of predicting answers as incoherent. Precision is lower than recall for both models. On the other hand, the uniform model achieves an F1-score of 30% in the test set. Therefore, we use it as a baseline for assessing the performance of other models.

The unsupervised model (rule-based) outperformed the supervised model (NB + BOW). The NB model depends heavily on the frequency of words in a bounded vocabulary, resulting in lower false negatives ($FN$) than false positives ($FP$) in the test set, as well as a 100% precision and 2.2% recall. The rule-based model is more robust, achieving a 57.7% F1-score. This is much higher than the 4.3% of the NB model. The rule-based model requires no training and obtains a similar performance in all three datasets, close to a 53.0% F1-score. Overall, these results demonstrate the superiority of the unsupervised model over the supervised model and suggest that unsupervised models may be better suited for dealing with different word domains.

Thus, the best baseline model for this study is the rule-based model. We now study the performance of the other models (Table 7).

### 4.2. Shallow Models

Table 7 shows that a model which mixed attributes (XGBoost + Mix) performed better than the rest of the shallow models. It had a 72.20% F1-score. The other models were the ones that used manually designed features (XGBoost + HF), and the one that used the default word vectors features obtained with the BETO language model (SVM + BETO embeddings). The difference in performance suggests that combining expert-designed features with word embedding features yields better results than either type of feature alone. Additionally, analysis of the shallow models revealed that the model using manually designed features (XGBoost + HF) performed better than the model using word vectors from the BETO language model (SVM + BETO embeddings). The difference in performance was likely due to the hand-engineering features being tailored to detect incoherence in answers to open-ended questions, while the BETO language model was trained on a general corpus of text which may not have captured the domain-specific structure.

### 4.3. Deep Models

Table 7 shows that the multi-tasking model outperformed the fine-tuning model. The first one with an F1 score of 79.1% on the test dataset and 84.3% on the validation dataset. They were better than the corresponding F1 score obtained with the fine-tuning model: 76.7% and 82.9%. The performances were much better than the best baseline model. Additionally, the model with multi-tasking had a better recall in the test set (82.3%) compared to fine-tuning (78.6%), while the precision was not significantly different between the two models. The difference in performance suggests that using a regularization factor, such as predicting the question type, can help avoid overfitting and improve generalization on different datasets.

Transfer of learning was significantly more robust for the deep learning models than the shallow ones. In a test set, the F1-scores of both deep models were considerably higher than those of the three shallow models. In particular, the deep models that used the embeddings of the BETO language model (SVM + BETO embeddings) outperformed the

shallow model when using the fine-tuning (and multi-tasking) strategy. The difference in performance suggests that transferring the learning of the default word vectors of the BETO model to the domain-specific structure of the open-ended question-answer pairs can significantly improve the performance of the BETO language model for predicting incoherence. Furthermore, the attention mechanism of the BERT architecture is likely to be responsible for the robustness and generalization capabilities of the deep models. It probably better captures the patterns between the question-answer pairs than manual feature estimators do. In the future, we plan to investigate which structures of open-ended answers deep models capture patterns to predict incoherence.

*4.4. Ensemble Models*

Table 7 shows that the general model outperformed the logical model in all three datasets. The F1-scores on the test set were 77.6% and 70.7%, respectively. The difference in performance indicates that the general architecture is better at learning the appropriate assembly, as well as being better able to cope with potential prediction errors of the individual classifiers. In contrast, the logical architecture had a precision of 66.4% and recall of 75.7%, indicating that it often produced more false positives than false negatives. This resulted in a higher proportion of incoherent answers.

The ensemble general architecture had a 77.6% F1-score on the test dataset. This was slightly less than the score of the BETO multi-tasking model. These were the two best-performing models. These results suggest that the use of an attention mechanism with additional task information can provide better results than relying on more traditional methods.

**5. Discussion**

Our study of the task of detecting incoherence in answers to open-ended mathematical questions for fourth graders revealed that it is a challenging task. We found that the best-performing baseline model was the unsupervised rule-based model, which achieved an F1-score of 57.76%. This was much better than the supervised NB model with bag-of-words, which obtained an F1-score of 4.32%. By combining representations provided by transformer architectures with handcrafted feature sets designed to address particular aspects of answers, we improved these results significantly. The shallow model that mixed features obtained an F1-score of 72.20%. These results are similar to those achieved by methods reported in the ASAG literature [28].

Recent research suggests that multi-task strategies, which consider a related secondary task, such as question type prediction, can improve the performance of deep models. The authors of [53] demonstrated this from experiments, which showed that multi-task training improved performance on unseen answers compared to a model without multi-task training. Studies such as [28,54,55] have also shown the benefits of taking the question into account for improved performance. We found that a multi-task training approach achieved a 79.1% F1-score. This was better than the 76.7% F1-score from fine-tuning and the 64.36% F1-score from a model using pre-trained word embeddings.

On the other hand, we found that ensemble models, which combine multiple classifiers according to the type of question, also have strong predictive power. They even show competitive performance against deep models. Previous studies, such as [56–58], have shown that ensemble models can achieve competitive performance on international semantic evaluation and reduce overfitting due to their heterogeneity. Our results also support the competitive performance of ensemble models, with an F1-score of 70.7% for the logical architecture and 77.6% for the general architecture. Moreover, the precision for the general architecture was the best obtained on the test dataset. This means that this ensemble model had the highest proportion of answers correctly predicted as incoherent between all the answers that the model predicted as incoherent. When this model predicted that an answer was incoherent, then with high probability it was incoherent.

In future work, we intend to propose a method for improving the performance of a deep learning model for detecting incoherent answers to open-ended fourth-grade mathematical questions. The proposed method will involve pretraining an architecture similar to BERT with fourth-grade mathematical vocabulary, following the example of MathBERT developed by [59]. To effectively train our version of the MathBERT model, we need to collect datasets that include fourth-grade mathematical language. Additionally, we will apply post hoc local and global interpretability techniques stratified by question type with surrogate models, as defined by [60]. Further research will focus on automatically detecting incoherence, building intrinsically explainable models in an educational context (XAI-ED) according to [61], and implementing future-oriented research methods to guide the process of argumentation to mathematical problems [62]. Automatic incoherence detection aims to improve the way students produce mathematics exercises on an online platform. Similarly, to predict end-of-year learning outcomes, attention should be paid to the concerns expressed by [63] about the use of written responses as a proxy indicator of negative affective states in learners [64]. Additionally, we plan to use these features to improve the prediction of students' long-term learning outcomes [65].

A recent study by [66] suggested that using machine-learning-enabled automated feedback can be an effective means of providing students with immediate feedback and support in the revision of their scientific arguments, as demonstrated by [67]. To further explore this idea, we plan to conduct a new study in which we would use NLP algorithms to automatically detect any inconsistencies or illogical statements in written responses to open-ended questions. Then, we could use the feedback generated by these algorithms to provide students with more targeted and specific guidance to help them revise their work and improve the overall quality of their reasoning.

## 6. Conclusions

One of the basic skills required in today's society is the ability to argue and justify decisions. This is also critical in mathematics. Being able to justify and explain shows a deeper understanding. Communicating with third parties is also central to collaborating with others and implementing solutions. Writing the arguments makes it easy for others to review the arguments. It also makes it possible to receive feedback from others. Although arguing in writing generates additional cognitive efforts to the calculation process, the evidence shows that it improves learning in the long term and facilitates a deeper understanding.

However, developing argumentative and communicative skills in primary classrooms is a great challenge. The vast majority of elementary school students are primarily trained in calculations and performing procedures. They are not used to giving justifications, much less writing them.

But learning to write explanations requires at least two conditions: all students must write and all receive immediate feedback. Thus, the challenge of teaching argumentation in mathematics is not simple. One solution is to use online platforms where students write their explanations. However, this is very demanding for the teacher. The teacher must review 30 answers in one or two minutes. To facilitate the review, it is necessary to automate the detection of incoherent responses. Thus, the teacher can immediately ask the student to correct or complement the answer. Some incoherent responses are obvious. Others are more complex. They require understanding and comparison with the question.

In this paper, we analyzed thousands of responses to open-ended questions written by fourth-grade students on the ConectaIdeas online platform. Using NLP and ML algorithms, we built several automatic classifiers to detect incoherent responses. We tested the classifiers on a completely independent dataset. These were other questions, posed in another school year, and answered by completely new fourth-grade students. We found two classifiers with very good performance. One used deep models, multi-layer neural networks, reaching an F1-score = 79.15% for incoherent detection. Another, used an initial classifier of the question type. This other classifier had a similar performance to the best classifier. However,

such a classifier has the advantage that, by its structure, it can help to understand what the classification is due to.

This research demonstrated that automatic classifiers could effectively detect incoherent answers to open-ended mathematics questions. However, this technique has limitations that warrant further research, such as exploring other methods of detecting incoherent written responses and investigating how we can use this technology in education. Automated detection of irrelevant written responses may also be a key component of ASAG systems. It can help to identify responses that are unrelated to the question or attempt to mislead the system. Additionally, a robust system with this capability has the potential to provide teachers with a useful tool to assess students' understanding of mathematical concepts, as well as reducing the amount of manual grading. Thus, teachers can focus on improving the development of key competencies [68], monitoring student progress [69], and improving long-term outcomes [65].

Finally, our study makes significant contributions to the mathematics education community. First, by analyzing thousands of responses to open-ended questions, we demonstrated the effectiveness of natural language processing and machine learning algorithms in detecting incoherent answers. Our classifiers outperformed systems that relied on different heuristics, highlighting the potential of these advanced techniques in educational assessment. Furthermore, our exploration of the use of an initial question-type classifier contributes to a deeper understanding of the classification process itself. These novel findings not only represent a breakthrough in the field of mathematics education, but also provide practical tools and approaches for improving student learning, assessment, and the development of crucial competencies, such as argumentative and communicative skills, in elementary classrooms.

**Author Contributions:** Conceptualization, R.A. and F.U.; methodology, R.A. and F.U.; software, F.U.; validation, R.A. and F.U.; formal analysis, R.A. and F.U.; investigation, R.A. and F.U.; resources, R.A.; data curation, F.U. ; writing—original draft preparation, F.U.; writing—review and editing, R.A. and F.U.; visualization, F.U.; supervision, R.A.; project administration, R.A.; funding acquisition, R.A. All authors have read and agreed to the published version of the manuscript.

**Funding:** This work was supported by the Chilean National Agency for Research and Development (ANID), grant number ANID/PIA/Basal Funds for Centers of Excellence FB0003.

**Institutional Review Board Statement:** Ethical review and approval were waived for this study, due to it being a class session during school time. The activity was revised and authorized by the respective teachers.

**Informed Consent Statement:** Student consent was waived due to authorization from teachers. Given that there were no patients but only students in normal sessions in their schools, within school hours, and using a platform that recorded their responses anonymously, the teachers authorized the use of anonymized information.

**Data Availability Statement:** Not available. Code available in public GitHub repository: https://github.com/furrutiav/coherence-prediction-systems-2023 (accessed on 17 April 2023).

**Acknowledgments:** Support from ANID/PIA/Basal Funds for Centers of Excellence FB0003 is gratefully acknowledged.

**Conflicts of Interest:** The authors declare no conflict of interest. The funder had no role in the design of the study; in the collection, analyses, or interpretation of data; in the writing of the manuscript; or in the decision to publish the results.

## Appendix A

*Appendix A.1. Baseline*

The most frequent model always predicts the class with the highest frequency in the dataset, which in our case is the coherent class (C0). The stratified model randomly predicts the incoherent class (C1) with a probability of $p = 0.13$ and the coherent class (C0) with

a probability of $1 - p$. The uniform model is similar to the stratified model but with a probability of $p = 0.5$.

The multinomial NB is a supervised learning method that seeks to find the class that maximizes the probability of a given text [52]. It utilizes Bayes' Theorem and assumes independence among the words of a text to calculate the probability of a text belonging to a particular class [38]. For incoherent classification, we use the NB model, which calculates the probability of a question-answer pair belonging to the incoherent class based on the proportion of incoherent and coherent texts and the proportion of words from these texts.

The rule-based model utilizes simple if/else rules to classify a question-answer pair. It has five rules, where the first rule ensures that the answer does not contain any of the keywords of question-independent incoherent answers. The remaining rules utilize keywords of three different types of questions to classify the pairs. The model identifies the incoherent class when the first rule is satisfied and at least one of the remaining rules is also satisfied.

The keywords of question-independent incoherence include curse words, recurring words, and representations of faces. The Q2-type questions have keyword pairs of the form $(k, k')$, where $k$ is the beginning of a quantity question, and $k'$ is a recurring word to ask for an explanation of the answer. The Q1-type questions have keywords from the beginning of quantity questions. The Q3-type questions have keyword pairs of the form $(k_1, k_2)$, where $k_1$ is the beginning of the question that asks for correctness, and $k_2$ is a recurring word for a binary-answer yes/no.

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
