# Peer review of "Automatically Detecting Incoherent Written Math Answers of Fourth-Graders"

_systems, doi:10.3390/systems11070353_

Round 1

Reviewer 1 Report

The topic of this study entitled “Automatically Detecting Incoherent Written Math Answers of Fourth-graders” is interesting and important. The paper is well written. Few comments are as follows: 

Abstract

The abstract is too lengthy. Especially, the beginning few sentences should be shorten (e. g., from “ Arguing and communicating is …..” to “Some incoherent answers are obvious.”).

 Introduction

The research questions should be presented in the paper. It would be better for the readers to understand the purposes of this study.

 Methods

The theoretical background of three categories: shallow models, deep 288 models, and ensemble models use in this study should be discussed in the literature review.

 Conclusion

What are the major contributions of this study to the community of mathematics education? Are there any differences of findings different from the earlier studies?

Author Response

Thank you for your time and dedication to reviewing our manuscript. We attached a file with a reply.

Reviewer 2 Report

In this manuscript, the authors propose an algorithm to detect incoherent written math answers of fourth-graders. Firstly, they collected a dataset which contains questions and answers generated in 2017 and 2019. Then, three kinds of methods are applied to classify the type of answers, shallow model, deep model, and ensemble model, respectively. There are some major issues in this manscript.

a. Dataset

1. They mentioned that three teachers and one member of the research team labeled all 2019 questions and answers. In a second stage, we reviewed the laberling, and we added the type of incoherence. Why the teachers did not join the second stage? 

2. The authors could provide more detail about this dataset. For example, some samples can be present in the manscript. Moreover, a correlation between the data in 2017 and the one in 2019 could be provided.

3. I suggest that the authors could make the dataset and source code public.

b. Methodolody

The authors describe the tasks in Sec. 3.0.3, they define four classifier functions. However, these classifier is closely related. They could design a framework to get all results from one model. 

Extensive editing of English language required

Author Response

(The authors gave the same response as above.)

Round 2

Reviewer 2 Report

The authors have been addressed my questions. Thus, I suggest to accept this manuscripts in current version.

The authors have been addressed my questions. Thus, I suggest to accept this manuscripts in current version.